# Characteristics and Outcome of Children with Wilms Tumor Requiring Intensive Care Admission in First Line Therapy

**DOI:** 10.3390/cancers14040943

**Published:** 2022-02-14

**Authors:** Anouk Steur, Paulien A. M. A. Raymakers-Janssen, Martin C. J. Kneyber, Sandra Dijkstra, Job B. M. van Woensel, Dick A. van Waardenburg, Cornelis P. van de Ven, Alida F. W. van der Steeg, Marc Wijnen, Marc R. Lilien, Ronald R. de Krijger, Harm van Tinteren, Annemieke S. Littooij, Geert O. Janssens, Annemarie M. L. Peek, Godelieve A. M. Tytgat, Annelies M. Mavinkurve-Groothuis, Martine van Grotel, Marry M. van den Heuvel-Eibrink, Roelie M. Wösten-van Asperen

**Affiliations:** 1Princess Máxima Center for Pediatric Oncology, 3584 CS Utrecht, The Netherlands; anouk.steur@gmail.com (A.S.); p.a.m.a.raymakers-janssen@umcutrecht.nl (P.A.M.A.R.-J.); c.p.vandeven-4@prinsesmaximacentrum.nl (C.P.v.d.V.); A.F.W.vanderSteeg@prinsesmaximacentrum.nl (A.F.W.v.d.S.); m.h.w.wijnen-5@prinsesmaximacentrum.nl (M.W.); r.r.dekrijger-2@prinsesmaximacentrum.nl (R.R.d.K.); h.vantinteren@prinsesmaximacentrum.nl (H.v.T.); g.o.r.janssens@umcutrecht.nl (G.O.J.); a.m.l.peek@prinsesmaximacentrum.nl (A.M.L.P.); g.a.m.tytgat@prinsesmaximacentrum.nl (G.A.M.T.); a.m.c.mavinkurve-groothuis@prinsesmaximacentrum.nl (A.M.M.-G.); M.vanGrotel@prinsesmaximacentrum.nl (M.v.G.); m.m.vandenheuvel-eibrink@prinsesmaximacentrum.nl (M.M.v.d.H.-E.); 2Department of Pediatric Intensive Care, Wilhelmina Children’s Hospital/University Medical Center Utrecht, 3584 EA Utrecht, The Netherlands; 3Division of Pediatric Critical Care Medicine, Beatrix Children’s Hospital/University Medical Center Groningen, 9713 GZ Groningen, The Netherlands; m.c.j.kneyber@umcg.nl (M.C.J.K.); s.k.dijkstra@umcg.nl (S.D.); 4Department of Pediatric Intensive Care, Amsterdam University Medical Centers, 1105 AZ Amsterdam, The Netherlands; j.b.vanwoensel@amsterdamumc.nl; 5Department of Pediatric Intensive Care, Maastricht University Medical Center, 6229 HX Maastricht, The Netherlands; d.van.waardenburg@mumc.nl; 6Department of Pediatric Nephrology, Wilhelmina Children’s Hospital/University Medical Center Utrecht, 3584 EA Utrecht, The Netherlands; m.lilien@umcutrecht.nl; 7Department of Pathology, University Medical Center Utrecht, 3584 CX Utrecht, The Netherlands; 8Department of Radiology, Wilhelmina Children’s Hospital/University Medical Center Utrecht, 3584 EA Utrecht, The Netherlands; A.S.Littooij-2@umcutrecht.nl

**Keywords:** Wilms tumor, intensive care, pediatric oncology, risk factors, outcome

## Abstract

**Simple Summary:**

Survival of children with Wilms tumor is excellent. However, treatment-related complications may occur, requiring treatment at the pediatric intensive care unit (PICU). The aim of our retrospective study was to assess the frequency, clinical characteristics, and outcome of 175 children with Wilms tumor requiring treatment at the PICU in the Netherlands. Thirty-three patients (almost 20%) required unplanned PICU admission during their disease course. Younger age at diagnosis, intensive chemotherapy regimens, and bilateral tumor surgery were risk factors for these unplanned PICU admissions. Three children required renal replacement therapy, two of which continued dialysis after PICU discharge. Two children died during their PICU stay. During follow up, hypertension and renal dysfunction were frequently observed, which justifies special attention for kidney function and blood pressure monitoring during and after treatment of these children.

**Abstract:**

Survival rates are excellent for children with Wilms tumor (WT), yet tumor and treatment-related complications may require pediatric intensive care unit (PICU) admission. We assessed the frequency, clinical characteristics, and outcome of children with WT requiring PICU admissions in a multicenter, retrospective study in the Netherlands. Admission reasons of unplanned PICU admissions were described in relation to treatment phase. Unplanned PICU admissions were compared to a control group of no or planned PICU admissions, with regard to patient characteristics and short and long term outcomes. In a multicenter cohort of 175 children with an underlying WT, 50 unplanned PICU admissions were registered in 33 patients. Reasons for admission were diverse and varied per treatment phase. Younger age at diagnosis, intensive chemotherapy regimens, and bilateral tumor surgery were observed in children with unplanned PICU admission versus the other WT patients. Three children required renal replacement therapy, two of which continued dialysis after PICU discharge (both with bilateral disease). Two children died during their PICU stay. During follow-up, hypertension and chronic kidney disease (18.2 vs. 4.2% and 15.2 vs. 0.7%) were more frequently observed in unplanned PICU admitted patients compared to the other patients. No significant differences in cardiac morbidity, relapse, or progression were observed. Almost 20% of children with WT required unplanned PICU admission, with young age and treatment intensity as potential risk factors. Hypertension and renal impairment were frequently observed in these patients, warranting special attention at presentation and during treatment and follow-up.

## 1. Introduction

Childhood renal tumors account for approximately 6% of all pediatric malignancies [1]. The majority of these tumors are Wilms tumors (WT). Although advances in treatment strategies have led to an excellent outcome for children with localized WT, future challenges lie in improving survival of specific subgroups and decreasing direct as well as and late toxic effects of treatment [2,3,4,5,6]. These treatment-related toxicities can require admission to the pediatric intensive care (PICU). However, the prevalence and risk factors of critical illness at presentation and during treatment, and its outcome have not been studied intensively. In general, children with cancer requiring PICU admission represent a highly complex and challenging group, with a survival inferior to that of the general PICU population [7]. Case reports confirm that children with WT may require unplanned intensive care treatment due to various tumor- and treatment-related emergencies. These cases revealed not only relatively well-known emergencies associated with WT at presentation, such as malignant hypertension and extensive (intracardial) tumor thrombus, but also postoperative hemorrhage, cardiomyopathy and hepatotoxicity, intracranial bleeding, and seizures [8,9,10,11,12,13,14,15,16,17,18].

In view of the scarce available data on PICU admissions of patients with WT, we conducted a retrospective, multicenter observational cohort study to describe the frequency, clinical characteristics, and outcomes of PICU admissions in children with an underlying WT in the Netherlands.

## 2. Materials and Methods

### 2.1. Study Design

We conducted a retrospective cohort study of Dutch patients diagnosed with WT and treated in one of the five participating Dutch tertiary care hospitals from January 2003 to January 2019. After January 2015, all cases were diagnosed and treated in the national pediatric oncology center, the Princess Máxima Center for Pediatric Oncology, in Utrecht, the Netherlands. We included only subjects with a primary WT, aged <18 years at diagnosis, who gave consent for registration in the International Pediatric Oncology Society Renal Tumour Study Group (SIOP-RTSG) Registry and/or the Princess Máxima Center Data & Biobank Registry. Patients with a non-WT renal tumor were excluded from the current analysis.

### 2.2. Data Collection

Baseline, diagnostic, treatment, and outcome data were collected for all included patients. Baseline characteristics included demographic variables such as gender and age at diagnosis. Disease stage was documented, in which metastasized WT patients at diagnosis were included in the stage IV group regardless of local abdominal stage. The histological diagnosis was classified as low, intermediate, or high risk according to the revised SIOP working classification of Renal Tumours of Childhood 2001 [19]. Diffuse anaplastic and blastemal predominant histological types were documented as high-risk WTs. The presence of a congenital or genetic anomaly was defined as the presence of either one or more congenital malformations and/or dysmorphic features, or a confirmed genetic germline aberration and/or confirmed syndrome. Isolated skin lesions, such as café-au-lait macules or solitary hemangiomas, were not classified as a congenital malformation in this study.

Details on treatment were retrieved from the medical records. Preoperative chemotherapy was classified as none (upfront tumor surgery), AV (4-week regimen containing vincristine and dactinomycin), AVD (6-week regimen containing vincristine, dactinomycin, and doxorubicin), or in exceptional cases >3 drugs (any regimen containing more than three chemotherapeutic agents). Surgical treatment variables included renal surgery type (tumor nephrectomy or nephron sparing surgery), tumor thrombectomy and/or metastectomy, performed as part of the first-line treatment, as described in the corresponding surgical report.

Postoperative chemotherapy regimens were classified as none, vincristine, AV-1 (≤5-week regimen containing vincristine and dactinomycin), AV-2 (27-week regimen containing vincristine and dactinomycin), AVD (regimen containing vincristine, dactinomycin, and doxorubicin) and >3 drugs (any regimen containing more than three chemotherapeutic agents). We documented whether postoperative radiotherapy was administered, as well as the applied radiotherapy dose and field.

For each PICU admission, additional data were collected, including time from WT diagnosis to PICU admission, treatment phase at PICU admission, indication for PICU admission, and pediatric index of mortality (PIM2) score at PICU admission [20]. The treatment phase at PICU admission was categorized into the following phases: diagnostic (prior to administration of any form of oncological treatment), preoperative chemotherapy, renal tumor surgery, postoperative treatment (chemotherapy and/or radiotherapy), and follow-up (≥1 week after the completion of WT treatment). Indications for PICU admission were categorized as postoperative monitoring, respiratory failure, circulatory failure, sepsis, neurological failure, kidney injury, or other.

### 2.3. Outcomes

The primary outcome was the frequency of unplanned PICU admissions during first line therapy and follow-up. A planned PICU admission was defined as postoperative monitoring following planned diagnostics under anesthesia or surgery. All other indications for PICU admission were deemed unplanned PICU admissions. Secondary outcomes included PICU length of stay (LOS), use of PICU resources (respiratory support, vasopressor and/or inotropic support, extracorporeal membrane oxygenation (ECMO), inhaled nitric oxide, and continuous renal replacement therapy (CRRT)), PICU mortality, overall mortality, disease outcome variables, and cardiac and renal morbidity at end of oncological treatment and last follow-up.

Disease outcome variables were relapse and progression (including site and subsequent treatment), death, or survival. In case of death, the following variables were recorded: cause of death (categorized as tumor-related, treatment-related, unrelated, or unknown) and time between WT diagnosis and death. The International Pediatric Oncology Mortality Classification Group (IPOMCG) definition of treatment-related mortality was used [21].

Cardiac morbidity was defined as left-ventricular shortening fraction < 28% confirmed by cardiac ultrasound and/or on active treatment by cardiologist for cardiomyopathy. Variables regarding renal morbidity were hypertension, renal function impairment, and use of renal replacement therapy (RRT) including renal transplantation. The presence of hypertension was defined as three consecutive blood pressure measurements ≥95th percentile for age according to the American Academy of Pediatrics clinical practice guideline and/or the use of antihypertensive medication [22]. If blood pressure measurements were not performed or documented, and the patients did not receive antihypertensive treatment, this was classified as absence of hypertension. The presence of chronic kidney disease (CKD) was defined according to the Kidney Disease: Improving Global Outcome (KDIGO) classification: i.e., estimated glomerular filtration rate (GFR) category < 90 mL/min/ 1.73 m^2^ and/or the presence of proteinuria, defined as an albumin/creatinine ratio ≥3 mg/mmol [23].

### 2.4. Statistical Analysis

The clinical characteristics and outcomes of WT patients with unplanned PICU admissions were compared to children without or with planned PICU admissions only. These latter two groups together represented the control group.

Categorical variables were displayed as frequencies (%) and compared using chi-square or Fisher exact test. Normality was tested with the Kolmogorov–Smirnov test. Continuous variables were displayed as medians with interquartile ranges (IQRs) and were compared with the Wilcoxon rank-sum test. Outcome of cardiac and renal morbidity at end of oncological treatment and last follow-up were compared with Jonckheere trend test. Data analysis was generated using SPSS version 25 (IBM, Armonk, NY, USA).

## 3. Results

### 3.1. Study Population

A total of 256 children with a renal tumor who were treated in one of the five participating tertiary care centers between 1 January 2003 and 1 January 2019 were registered in the SIOP-RTSG NL Registry and/or the Princess Máxima Center Data & Biobank Registry. Of those, 69 children were excluded because of a histological diagnosis not concordant with WT, and four were excluded because of an age older than 18 years at diagnosis. For six patients, there was insufficient information available to allow medical chart review. Two patients were excluded as they had been admitted to PICU only during relapse treatment leaving a total of 175 included patients (Figure 1).

Seventy-eight WT patients (45%) were admitted at least once to PICU during their treatment, resulting in 120 PICU admissions in this patient group, including 50 unplanned PICU admissions (42%) and 70 planned PICU admissions (58%).

### 3.2. Characteristics of Patients Requiring PICU Admission

Baseline characteristics of patients requiring unplanned PICU admissions versus the control group are described in Table 1. Thirty-three children with WT required unplanned PICU admission during the course of first-line treatment and follow-up (19%). Patients with an unplanned PICU admission were significantly younger, had received more intensive chemotherapy regimens, and presented more often with bilateral tumors when compared to patients with only planned or no PICU admissions (control group) (Table 1). No significant differences were found in other disease stage, metastases, and histology between patients with unplanned PICU admissions and the control group.

### 3.3. Indications for Unplanned PICU Admissions per Treatment Phase

Of the 50 unplanned PICU admissions, in 33 unique patients (nine patients were admitted twice and four patients were admitted three times), 12 occurred in the diagnostic phase (11 patients), 6 during pre-operative chemotherapy (6 patients), 15 after renal tumor surgery (14 patients), 16 during the postoperative treatment phase (11 patients), and 1 during follow-up (1 patient) (Figure 2 and Table 2). Indications for these admissions consisted primarily of respiratory and/or circulatory failure in all treatment phases.

Causes for respiratory and circulatory failure in both the diagnostic and preoperative chemotherapy phase were mainly tumor-related (Table 2). Respiratory failure was mainly due to the presence of abdominal tumor mass, with or without pulmonary metastases and accompanying pulmonary infection. Hypertension was the main cause of circulatory failure. Other causes were intra-tumor hemorrhage and one case of circulatory failure secondary to gastrointestinal obstruction caused by Wilms tumor mass. No other hemorrhages were observed. Information of paraneoplastic Von Willebrand Factor was not documented.

The majority of unplanned PICU admissions following renal tumor surgery were due to circulatory failure, most frequently caused by intraoperative abdominal hemorrhage and hypotension. PICU admission indications in the postoperative treatment phase were more diverse and mainly treatment-related. In the one unplanned PICU admission during follow-up 34 months after discontinuation of WT treatment, the admission indication was neurological complications following preexisting scoliosis correction, hence unrelated to WT.

### 3.4. PICU Stay Characteristics

The median PIM2 score was 1.0 (IQR 0.5–1.38; range 0.1–18.9) for unplanned PICU admissions (range 0.1–18.9) compared to 0.25 (IQR 0.2–0.57; range 0.1–2.4) in the control group (Table 3). WT patients with an unplanned PICU admission had a median PICU LOS of 3.0 days (IQR 1.0–6.0), in contrast to a median PICU LOS of 1.0 day (IQR 1.0–1.0) observed in planned PICU admissions. Three patients with an unplanned PICU admission had a LOS exceeding 20 days, reflecting a complicated PICU course. This included one infant, and two children with advanced stage disease requiring intensive cancer treatment regimens. The first, a 2-month-old girl with stage I WT, was admitted for respiratory insufficiency secondary to abdominal tumor mass with chyloperitonitis requiring tumor nephrectomy, with a PICU course complicated by postoperative chyloperitonitis, hypertension, acute kidney injury, syndrome of inappropriate ADH secretion (SIADH), Klebsiella pyelonephritis, and invasive candidiasis. The second, an 18-month-old girl with stage IV WT, was admitted for respiratory insufficiency secondary to combined influenza A and post-irradiation ARDS, requiring high-frequency oscillation (HFO), inhaled NO, inotropic support, and prolonged ventilatory support. The third, a 4-year-old boy with stage V high risk WT, was admitted for respiratory insufficiency secondary to neutropenic sepsis and concurrent end-stage kidney disease (ESKD), requiring 91 days of invasive ventilatory support, tracheostomy, and renal replacement therapy. The PICU admission was complicated by a tension pneumothorax, an intrathoracic hematoma requiring surgical intervention, S. Aureus sepsis, and invasive candidiasis.

PICU resource use was significantly higher in the WT patients with an unplanned PICU admission compared to planned admissions. Twenty-five unplanned PICU cases (50%) required invasive mechanical ventilation, 14 cases (28%) required vasopressor and/or inotropic support, and four cases (8%) needed continuous renal replacement therapy. During three admissions, occurring in two bilateral WT patients, continued hemodialysis and peritoneal dialysis were needed, respectively, at PICU discharge because of tumor and treatment-related ESKD, of which one patient had germline WT1 mutation, without glomerulosclerosis in the remnant normal kidney tissue compartment.

Two children died during PICU stay. The first, a five-year-old boy receiving postoperative treatment for bilateral WT while on peritoneal dialysis for ESKD, died from circulatory insufficiency secondary to massive bowel ischemia attributed to invasive candidiasis with immune reconstitution inflammatory syndrome. The second, a three-year-old girl, died during postoperative treatment for lung and bone metastasized WT, local stage III. She died from increased intracranial pressure due to a concomitant leptomeningeal rhabdomyosarcoma, which was identified only during treatment as a secondary malignancy [24].

### 3.5. Outcome of WT Patients

Outcomes of the patients are depicted in Table 4. Median follow-up time was 52 months (IQR 30–115 months). At last follow-up, hypertension and renal function impairment CKD stage ≥ 2 was found to be more frequent in the survivors of unplanned PICU admitted patients when compared to the control group (*p* = 0.010 and *p* < 0.001, respectively). Five patients revealed CKD stage ≥3 renal function impairment after unplanned PICU admission, compared to no patient in the control group. Of these five patients, two developed ESKD. No differences were found for cardiac morbidity between patients after unplanned PICU admission and the control group. In addition, no significant differences of occurrence of relapse/refractory disease were found between both groups. Death was reported in six of 175 (3.4%) patients. Three patients died in the control group, due to tumor-related causes. Of the three patients that died in the unplanned PICU admission group, two children died during PICU stay from treatment-related toxicity and a secondary malignancy, respectively, and one patient had a tumor-related death during follow-up (relapsed WT).

## 4. Discussion

To the best of our knowledge, this is the first multi-center cohort study describing the frequency, admission indications, and outcome of Wilms tumor patients requiring PICU admission during the course of the disease. In general, current treatment strategies result in excellent survival rates for most children with WT, and therefore, gaining insight into critical events is essential. Our study shows that almost 20% of children with WT require unplanned PICU admission during the course of first-line treatment and follow-up, of which one-third is before the start of any treatment as a result of tumor-related complications. Although 50% of patients with unplanned PICU admissions require invasive ventilatory support, most unplanned PICU admissions in this cohort had a relatively uncomplicated course, reflected in a short PICU length of stay, short ventilation duration, and short duration of inotropic support.

Interestingly, already during the diagnostic and preoperative chemotherapy treatment phase, serious tumor-related complications led to unplanned PICU admissions, with hypertension and respiratory failure due to large abdominal mass as the most prevalent causes of unplanned PICU admission. Hypertension is mainly based on activation of the renin-angiotensin aldosterone system (RAAS) by renin overproduction, in WT caused by local kidney ischemia due to vascular compression by large tumors, as well as directly by the blastemal component of the WT [25,26]. Untreated hypertension increases the risk of complications like cardiac decompensation, neurological deficits, and bleeding before or during renal surgery [26,27,28]. Whether a relatively short duration of tumor-related hypertension in WT patients may lead to long term adverse outcomes has not been sufficiently studied as yet. Previous studies have shown that more than half of WT patients suffer from hypertension at diagnosis, yet only 60% of these patients receive antihypertensive treatment [26,29]. The majority of children that received antihypertensive treatment were normotensive prior to surgery [26,29]. In our population, 4% (7/175) of children with WT required PICU admission for hypertension, with three of these seven during the preoperative chemotherapy phase. Potentially, early identification and treatment of hypertension with ACE inhibition may prevent the need for PICU admission and decrease the risk of hypertension-related complications. However, ACE-inhibition may also cause acute kidney injury in a kidney with diminished arterial flow due to compression caused by a tumor.

In the later treatment phase, PICU admission reasons were more diverse. In the general population of children with cancer requiring PICU admission, 12–38% of PICU admissions are due to sepsis, with a 50% PICU mortality rate [30,31]. In our cohort, there were few infectious complications leading to unplanned PICU admission, reflecting the generally lower intensity and consequent hematological toxicity of most WT treatment regimens when compared to the regimens required for, e.g., hematological malignancies. This may in part explain the low PICU mortality of 4% in this cohort, as compared to a 28% PICU mortality rate that has been reported in pooled cohorts of pediatric cancer patients [7].

Our study suggests that young and high-risk WT patients who receive intensive treatment regimens (i.e., for bilateral WT) seem to require unplanned PICU admission more often. This finding has not been described previously and is not consistently reflected in the previously reported cases of WT patients with complications requiring PICU admission [8,9,10,11,12,13,14,15,16,17,18]. In the SIOP93-01 and SIOP2001 studies, patients aged 6 months to 2 years were shown to have an excellent survival, with a higher event-free survival when compared to older age groups [2,3,32]. However, the results of our study show that, despite their superior survival outcome, young WT patients may carry a higher risk for treatment- and tumor-related complications leading to unplanned PICU admission in first-line treatment. It is conceivable that infants are more vulnerable to side effects of cytotoxic treatment. For that reason, dose adjustments are included in upfront treatment protocols [6]. Historically, excess complications and toxic deaths in infants treated in the first two National Wilms Tumor Study Group (NWTS) trials warranted a 50% chemotherapy dose reduction [33,34]. Similarly, following the International Society of Pediatric Oncology (SIOP) 6 study, unacceptable toxicity in infants led to a recommended chemotherapy reduction to 2/3 of the dose in infants and children with a body weight under 12 kg treated according to SIOP protocols [35]. Nevertheless, international consensus on how to adjust dosages in young children across all tumor types, based on pharmacokinetic research findings, is still pending. With regard to surgical complications, available evidence shows that surgical complications after primary nephrectomy for Wilms tumor are rare, and age was not found as a risk factor [36]. In line with these findings, of the 14 patients who were admitted to PICU due to post-operative complications, only four were under two years of age.

Children with bilateral WT tumors are particularly at risk for critical illness, based on often intensive and prolonged treatment, as well as because of the requirement of resection of a considerable proportion of the healthy renal parenchyma. In addition, in a proportion of children, kidney disease caused by germline *WT1* mutations can intrinsically already contribute to organ failure [37].

Despite the relatively uncomplicated admission course of the majority of unplanned PICU admissions in this cohort, a strikingly higher frequency of hypertension and chronic kidney disease was found at end of treatment and last follow-up in these patients compared to the control group. Chronic kidney disease stage ≥3 renal impairment was exclusively seen in unplanned PICU admitted patients, with ESKD occurring in two patients, both with bilateral WT, of which one had a *WT1* gene variant. Hypertension and renal function impairment are well-known long-term morbidities following WT treatment [38,39]. Known risk factors include nephrectomy, bilateral WT, specific cytotoxic agents such as platinum-based and alkylating agents, and radiation therapy involving the kidney region [40,41]. Whether complications warranting PICU admission and the subsequent PICU treatment are also risk factors for adverse renal outcomes, or whether this is related to the clinical and disease characteristics of the WT patients in this group, remains the question.

Surprisingly, two patients presented with post-extubation subglottic stenosis after Wilms tumor surgery. One patient, who was born at 33 weeks, presented with an antenatal WT. This patient was intubated immediately after birth. The post-extubation subglottic stenosis was probably due to an intubation trauma. The second patient was 3 years old, with no reported pre-operative respiratory complaints. After the planned surgery, the patient was respiratory insufficient after extubation. This turned out to be due to subglottic stenosis. A cause of this stenosis has not been found. We do not expect it is a complication of the vincristine administration because pre-operative, no signs of upper respiratory tract obstruction were noted.

This study is the first to describe PICU admissions in a reasonably sized, unbiased cohort of WT patients at five tertiary pediatric oncology centers. However, we also acknowledge several limitations. An inherent limitation of this study is its retrospective nature, and the fact that we rely on data that were collected from patients’ medical records primarily captured for clinical care and not for research. This descriptive cohort study is insufficiently powered to allow multivariable regression analyses to model risk factors for unplanned PICU admission and outcome. Also, further studies are required to determine whether the higher frequency of hypertension at diagnosis as well as renal function impairment observed in WT patients requiring unplanned PICU admission is the result of ICU treatment complications or related to patient characteristics requiring more intensive treatment or predisposing to renal toxicity. This study was not designed to create general guidelines, as local resource issues and indications for PICU admission may vary between different countries and may depend on variation in general/ environmental circumstances over time, such as the recent COVID pandemic.

Our study on critical illness in children with WT illustrates the importance of multidisciplinary specialized expert-centered care for children with renal tumors, as pointed out in the ExPO-R-Net initiative [6]. Accordingly, in the Netherlands, children with WT have been treated in specialized pediatric oncology centers over the last decades, and now, since November 2014, in only one dedicated national pediatric oncology center, by which we intend to increase survival and to increase quality of cure. Other initiatives will need to focus on more targeted effective, and less toxic, treatment modalities, to ultimately improve outcome by enhanced efficacy as well as decrease of treatment-related complications, especially in very young and intensively treated high-risk children.

## 5. Conclusions

In this multicenter study, with more than 15 years of inclusion, we observed that almost 20% of children with Wilms tumor required unplanned PICU admission, with young age and treatment intensity as potential risk factors. Hypertension and CKD stage 2 or higher was more frequently observed in unplanned PICU admitted patients, which justifies special attention for kidney function, development of proteinuria, and blood pressure monitoring during and after treatment of these children. Larger cohort studies are required to identify independent determinants of PICU admission and to clarify the role of events during PICU admission as a risk factor for hypertension and CKD as late morbidity in children with WT.

## Figures and Tables

**Figure 1 cancers-14-00943-f001:**
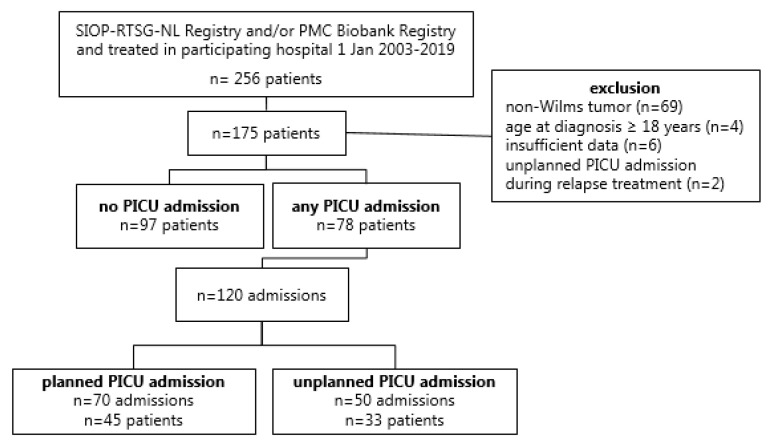
Number of PICU admissions in the study population. Abbreviations: SIOP-RTSG NL Registry: International Society of Paediatric Oncology Renal Tumour Study group–Dutch Registry; PMC: Princess Máxima Center for Pediatric Oncology; PICU: pediatric intensive care unit.

**Figure 2 cancers-14-00943-f002:**
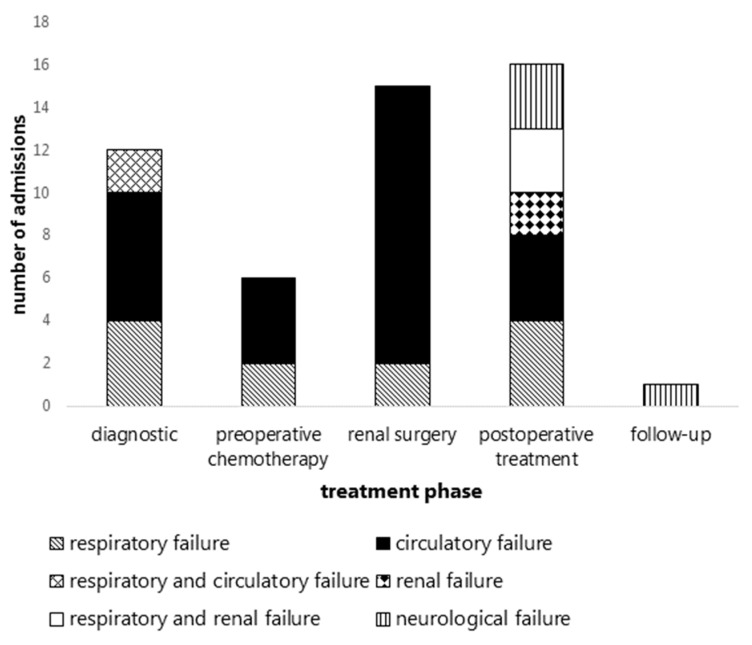
Indications for unplanned PICU admissions in children with WT per treatment phase.

**Table 1 cancers-14-00943-t001:** Baseline characteristics.

	Total Population *n* = 175 Patients	Control Group (No PICU Admission or Planned PICU Admission Only) *n* = 142 Patients	Unplanned PICU Admission *n* = 33 Patients
Gender, male, *n* (%)	82 (46.9)	70 (49.3)	12 (36.4)
Age in months, median (IQR)	38.5 (22.0–57.0)	42.2 (27.4–61.1)	22.0 (13.8–38.8) †
**Disease stage, *n* (%)**			
I	60 (34.3)	53 (37.3)	7 (21.2)
II	35 (20.0)	32 (22.5)	3 (9.1)
III	38 (21.7)	27 (19.0)	11 (33.3)
IV *	24 (13.7)	18 (12.7)	6 (18.2)
V	18 (10.2)	12 (8.5)	6 (18.2)
** abdominal stage*		*I (n = 3), II (n = 3), III (n = 11), V (n = 1)*	*I (n = 1), II (n= 0), III (n = 3), V (n = 2)*
**Metastases, *n* (%)**			
Lung	17 (9.6)	13 (9.2)	4 (11.4)
Liver	0 (0.0)	0 (0.0)	0 (0.0)
Other site	0 (0.0)	0 (0.0)	0 (0.0)
Multiple sites	7 (4.1)	5 (3.5)	2 (5.8)
**Histology, *n* (%)**			
Low risk	9 (5.1)	6 (4.2)	3 (9.1)
Intermediate risk	147 (84.0)	120 (84.5)	27 (81.8)
High risk	19 (10.8)	16 (11.3)	3 (9.1)
-Diffuse anaplasia	7 (4.0)	6 (4.2)	1 (3.0)
-Blastemal predominant	10 (5.7)	8 (5.6)	2 (6.1)
-Diffuse anaplasia and Blasternal predominant	2 (1.1)	2 (1.4)	0 (0.0)
**Documented genetic anomaly** **or congenital malformation, *n* (%) ****			
Beckwith-Wiedeman syndrome	5 (2.9)	5 (3.5)	0 (0.0)
Hemihypertrophy NOS	9 (5.1)	7 (4.9)	2 (6.1)
*WT1* gene variant	8 (4.6)	4 (2.8)	4 (12.1)
Genito-urinary malformation NOS	3 (1.7)	3 (2.1)	0 (0.0)
Other	17 (9.7)	11 (7.7)	6 (18.2)
**Preoperative treatment, *n* (%)**			
None	8 (4.6)	6 (4.2)	2 (6.1)
AV	138 (78.8)	116 (82.0)	22 (66.7)
AVD	24 (13.7)	18 (13.4)	6 (18.2)
>3 drugs	5 (2.9)	2 (1.4)	3 (9.1) †
**Renal surgery, *n* (%)**			
Nilateral	156 (89.1)	131 (92.2)	25 (75.7) †
Bilateral	19 (10.8)	11 (7.7)	8 (24.2) †
**Thrombectomy, *n* (%)**	10 (5.8)	7 (4.9)	3 (9.1)
**Metastectomy, *n* (%)**	6 (3.4)	4 (2.8)	2 (6.1)
**Postoperative treatment, *n* (%)**			
None	5 (2.9)	4 (2.8)	1 (3.0)
VCR	3 (1.7)	2 (1.4)	1 (3.0)
AV-1	41 (23.4)	38 (26.8)	3 (9.1) †
AV-2	57 (32.6)	43 (30.3)	14 (42.4)
AVD	38 (21.7)	33 (23.2)	5 (15.2) †
>3 drugs	31 (17.7)	22 (15.5)	9 (27.3) †
**Radiotherapy, *n* (%)**			
Abdominal	53 (30.3)	41 (28.9)	12 (36.4)
Pulmonary	2 (1.1)	2 (1.4)	0 (0)
Combined	6 (3.4)	3 (2.1)	3 (9.1)

** Other congenital or genetic anomalies, specified: no PICU admission or planned PICU admission only: 47XXY, neurofibromatosis type 1, cystinuria, heterozygous CHEK2/MUTYH mutation, familial paraganglioma, atrial septum defect, developmental delay NOS (*n* = 2), dysmorphic features NOS, ventricular septal defect with subvalvular stenosis and dysmorphic features NOS *) syndrome diagnosis was based on clinical assessment and not systematic counseling and NGS in those days unplanned PICU admission: Fanconi anemia (homozygous BRCA2-mutation), tuberous sclerosis (TSC1 mutation), chromosome 16q deletion, skeletal dysplasia NOS, antenatal WT, prematurity with macroglossia and developmental delay. IQR: interquartile range, WT: Wilms tumor, NOS: not otherwise specified, AV: dactinomycin/vincristine regimen, AVD: dactinomycin/vincristine/doxorubicin regimen, AV-1: ≤5 wk regimen dactinomycin/vincristine, AV-2: >5 week regimen dactinomycin/vincristine. † statistically significant difference *p* < 0.05.

**Table 2 cancers-14-00943-t002:** Indications for unplanned PICU admissions in children with WT per treatment phase-admission indications and underlying cause per individual unplanned PICU admission.

Clinical Characteristics (Gender, Age, Stage, Histology, History)	PICU Admission Indication	Causes Underlying PICU Admission Indication
**Diagnostic phase**		
F, 6 mo, I, IR	circulatory failure	hypertension
M, 1.6 y, I, IR	circulatory failure	intratumoral hemorrhage
F, 6 mo, I, IR	circulatory failure	hypertension
F, 1.3 y, II, IR, Fanconi anemia	respiratory and circulatory failure	ARDS (suspected infection), abdominal tumor mass, hypertension
F, 1.6 y, II, IR	circulatory failure	hypertension
M, 6 mo, III, LR *	circulatory failure	hypertension
F, 1.8 y, III, IR *	circulatory failure	intratumoral hemorrhage
F, 0 mo, III, IR, antenatal WT *	respiratory and circulatory failure	antenatal abdominal tumor mass
(2 PICU admissions)	respiratory failure	subglottic stenosis (post-extubation), laryngomalacia, omphalitis
F, 3.3 y, IV (lung, bone), IR *	respiratory failure	pulmonary metastases, suspected pneumonia
F, 1.5 y, IV (lung), IR *	respiratory failure	abdominal tumor mass, pulmonary metastases
F, 2.3 y, IV (lung), bilateral, IR	respiratory failure	abdominal tumor mass, pulmonary metastases, pleural effusion
**Preoperative chemotherapy phase**		
F, 2 mo, I, IR	respiratory failure	abdominal mass with chyloperitoneum
M, 6 mo, III, LR *	circulatory failure	hypertension
F, 1.8 y, III, IR *	circulatory failure	abdominal tumor mass causing bowel obstruction
F, 2.5 y, III, IR	circulatory failure	hypertension
F, 3.7 y, III, IR *	circulatory failure	hypertension
M, 9 mo, V, IR, *WT1* mutation	respiratory failure	abdominal tumor mass
**Renal tumor surgery**		
F, 1.5 y, I, LR	circulatory failure	left TN: transection SMA
F, 1.7 y, I, IR, hemihypertrophy **	circulatory failure	left NSS: intraoperative hemorrhage
(2 PICU admissions)	circulatory failure	left TN: hypotension
M, 4.3 y, II, IR, prematurity GA 26 wk	respiratory failure	left TN: inability to wean postoperatively, BPD
M, 2.8 y, III, IR	circulatory failure	right TN, thrombectomy VCI / RA: hypotension
M, 1.5 y, III, IR, *WT1* mutation	circulatory failure	right TN: intraoperative hemorrhage
F, 3.7 y, III, IR *	circulatory failure	right TN, partial hepatectomy, hemicolectomy: hypotension
M, 2.9 y, III, IR *	circulatory failure	right TN: intraoperative hemorrhage
F, 3.0 y, III, IR **	respiratory failure	right TN, partial hepatectomy: post-extubation subglottic stenosis
M, 2.9 y, III, IR, chr16 q deletion	circulatory failure	left TN: transection SMA
F, 7.1 y, III, HR (BP) *	circulatory failure	left TN, thrombectomy VCI: intraoperative hemorrhage
F, 3.2 y, IV (lung), IR, skeletal dysplasia *	circulatory failure	left TN: hypotension
M, 3.8 y, V, IR	circulatory failure	right TN and left NSS, day + 2: intra-abdominal urine leakage
F, 11 mo, V, IR, *WT1* mutation	circulatory failure	left TN, right tumor biopsy: hypotension
F, 5.1 y, V, IR, *TSC1* mutation	circulatory failure	left NSS: intraoperative hemorrhage
**Postoperative chemotherapy phase**		
F, 1.8 y, I, LR	circulatory failure	urosepsis (*Citrobacter* spp.)
F, 1.7 y, I, IR, hemihypertrophy **	respiratory failure, kidney injury	kidney injury (acute on chronic) with fluid overload, ascites, suspected infection
M, 2.9 y, III, IR *	respiratory failure	pleural empyema
F, 3.0 y, III, IR **	respiratory failure	subglottic stenosis (recurrence post-extubation stenosis)
(2 PICU admissions)	respiratory failure	subglottic stenosis (recurrence post-extubation stenosis)
F, 7.1 y, III, HR (BP) *	neurological failure	intracranial hemorrhage (thrombocytopenia)
F, 3.3 y, IV (lung, bone), IR *	neurological failure	raised ICP leptomeningeal secondary malignancy
F, 1.5 y, IV (lung), IR *	respiratory failure	influenza A and post-irradiation ARDS
F, 4.9 y, IV (lung, liver), HR (DA)	circulatory failure	sepsis, chylothorax, chyloperitoneum
M, 1.8 y, IV (lung), bilateral, IR **	circulatory failure	cardiomyopathy
(3 PICU admissions)	kidney injury	prerenal kidney injury (vomiting, gastric ulcer) in setting of CKD
	kidney injury	prerenal kidney injury (vomiting) in setting of CKD
F, 7 mo, V, IR, *WT1* mutation	respiratory failure, kidney injury	suspected viral airway infection, ESDR with fluid overload
M, 4.3 y, V, HR (BP), hemihypertrophy **	respiratory failure, kidney injury	ESKD with fluid overload, neutropenic sepsis
(3 PICU admissions)	neurological failure	seizures secondary to malignant hypertension and hypokalemia
	circulatory failure	massive bowel ischemia in setting of IRIS following invasive candidiasis
**Follow-up phase**		
F, 3.2 y, IV (lung), IR, skeletal dysplasia *	neurological failure	scoliosis correction: loss of neuromonitoring

* Total of 2 unplanned PICU admissions during the disease course (*n* = 9 patients). ** Total of 3 unplanned PICU admissions during the disease course (*n* = 4 patients). PICU: pediatric intensive care unit, F: female, M: male, y: years, mo: months, LR: low risk, IR: intermediate risk, HR: high risk, BP: blastemal predominant, DA: diffuse anaplasia, WT: Wilms tumor, TSC1: tuberous sclerosis complex 1, GA: gestational age, ARDS: acute respiratory distress syndrome, TN: tumor nephrectomy, NSS: nephron sparing surgery, SMA: superior mesenteric artery, BPD: bronchopulmonary disease, IVC: inferior vena cava, RA: right atrium, ICP: intracranial pressure, CKD: chronic kidney disease, ESKD: end-stage kidney disease, IRIS: immune reconstitution inflammatory syndrome.

**Table 3 cancers-14-00943-t003:** Characteristics of PICU admissions in children with WT.

	*Planned PICU Admissions* *(n = 70 Admissions)*	*Unplanned PICU Admissions* *(n = 50 Admissions)*
**Length of stay**		
LOS in days, median (IQR)	1.0 (1.0–1.0)	3.0 (1.0–6.0)
LOS in days, range	<1–6	<1–100
**PIM2 score ***		
PIM2 score, mortality probability %, median (IQR)	0.25 (0.2–0.57)	1.0 (0.5–1.38)
PIM2 score, mortality probability %, range	0.1–2.4	0.1–18.9
**Invasive ventilation, *n* (%)**	23 (32.9)	25 (50.0)
Days on invasive ventilation, median (IQR)	0.1 (0.1–0.2)	4.0 (1.0–1.5)
Days on invasive ventilation, range	0.1–2	<1–91
**Vasopressor/inotropic support, *n* (%)**	1 (1.4)	14 (28.0)
Days on vasopressor/inotropic support, median (IQR)	(-)	1.0 (0.3–1.0)
Days on vasopressor/inotropic support, range	<1	<1–4
**ECMO, *n* (%)**	0	0
**iNO, *n* (%)**	0	1 (2.0)
**Renal replacement therapy, *n* (%)**	2 (2.8)	4 (8.0)
Days on renal replacement therapy, range	2	2–47
Renal replacement therapy at discharge, *n* (%)	2 (2.8)	3 (9.1)
**PICU mortality, *n* (%)**	0	2 (4.0)

* Insufficient data to compute PIM2 in 17 admissions. LOS: length of stay, PIM: Pediatric Index of Mortality, IQR: interquartile range, ECMO: extracorporeal membrane oxygenation, iNO: inhaled nitric oxide.

**Table 4 cancers-14-00943-t004:** Outcome of children with WT.

	Total Cohort (*n* = 175 Patients)	Control Group (No or Planned PICU Admissions Only) (*n*= 142 Patients)	Unplanned PICU Admission (*n* = 33 Patients)
**Follow-up time in months, median (IQR)**	52 (30–115)	52 (31–124)	50 (21–85)
**Hypertension, *n* (%)** At end of treatment At last follow-up	13 (7.4) 12 (6.9)	7 (4.9) 6 (4.2)	6 (18.2) † 6 (18.2) †
**Impaired renal function, *n* (%)** Any at end of treatment CKD stage ≥2 at end of treatment CKD stage ≥3 at end of treatment Any at last follow-up CKD stage ≥2 at last follow-up CKD stage ≥3 at last follow-up	6 (3.4) 6 (3.4) 4 (2.3) 15 (8.6) 12 (8.9) 5 (2.8)	1 (0.7) 1 (0.7) 0 (0) 10 (7.0) 7 (4.9) 0 (0)	5 (15.2) † 5 (15.2) † 4 (12.1) † 5 (15.2) † 5 (15.2) † 5 (15.2) †
**Dialysis, *n* (%)** At end of treatment At last follow-up	2 (1.1) 1 (0.6)	0 (0) 0 (0)	2 (6.1) 1 (3.0)
**Kidney transplant, *n* (%)**	1 (0.6)	0 (0)	1 (3.0)
**Cardiomyopathy, *n* (%)** At end of treatment At last follow-up	2 (1.1) 3 (1.7)	1 (0.7) 2 (1.4)	1 (3.0) 1 (3.0)
**Relapse/Refractory WT, *n* (%)** Relapsed WT Refractory WT Time to relapse/progression in months, median (IQR)	11 (6.3) 4 (2.3) 16.0 (8–30.5)	8 (5.6) 3 (2.1) 15.5 (6.5–27)	3 (9.1) 1 (3.0) 30.5 (8.5–30.5)
**Overall mortality, *n* (%)** Time to death in months, median (IQR) Cause of death, *n* (%) - Tumor-related - Treatment-related - Secondary malignancy	6 (3.4) 20.5 (8–26.5) 4 (2.3) 1 (0.6) 1 (0.6)	3 (2.1) 26.5 (21.5–26.5) 3 (2.1) 0 (0) 0 (0)	3 (9.1) † 9.5 (6.5–9.5) 1 (3.0) 1 (3.0) 1 (3.0)

PICU: pediatric intensive care unit, IQR: interquartile range, WT: Wilms tumor, CKD stage: chronic kidney disease according to KDIGO: Kidney Disease–Improving Global Outcomes classification, GFR: glomerular filtration rate, † statistically significant difference *p* < 0.05.

## Data Availability

The data underlying this article may be shared on request to the corresponding author.

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
