# Peer review of "Characteristics and Outcome of Children with Wilms Tumor Requiring Intensive Care Admission in First Line Therapy"

_cancers, 2022, doi:10.3390/cancers14040943_

Round 1

Reviewer 1 Report

  1. What is the main question addressed by the research?

Wilms tumor is the most frequent renal tumor in childhood and the overall survival is excellent. For that reason, it is very important to know the possible complications related to the tumor or its treatment. The aim of this study is to assess the frequency, clinical characteristics and outcome of children with Wilms tumor requiring unplanned PICU admission during their disease course.

  1. Do you consider the topic original or relevant in the field, and if

so, why?

Yes, I do. As these are patients with a very good prognosis, it is interesting to analyze the causes of PICU admission and their risk factors.

  1. What does it add to the subject area compared with other published

material?

To my knowledge, there is no other published article on this topic.

  1. What specific improvements could the authors consider regarding the

methodology?

I think the methodology is correct.

  1. Are the conclusions consistent with the evidence and arguments

presented and do they address the main question posed?

Although the number of patients is limited, since it is a rare disease, I think it is an adequate number of patients and the authors made an appropriate discussion analyzing risk factors.

  1. Are the references appropriate?

Yes, the references are appropriate and updated.

  1. Please include any additional comments on the tables and figures.

 The tables and figures are clear, very complete, easy to understand and summarize in an adequate way.

 I think it is an interesting manuscript that provides greater knowledge about the causes of PICU admission and risk factors in patients with Wilms tumor.

I would just like to add a comment:

On page 3, line 92: “standard, intermediate or high risk”

It would be better: “low, intermediate or high risk”

Reviewer 2 Report

This is a multi-center study detailing the clinical characteristics and outcomes of Wilms tumor patients who require unplanned intensive care admission during first line therapy. I commend the authors on a very interesting study question, as critical care-focused pediatric oncology questions (at least for Wilms tumor) are quite rare in the literature. The study is well-written, reasonably concise, and the conclusions are supported by the data. 

I have the following suggestions for improvement of the manuscript prior to publication:

Major: 

  1. Often the criteria for ICU admission are quite different depending on the institution, and as we've seen in the pandemic, local resource issues. Therefore, this study may lack generalizability. This should be noted in the limitations paragraph.
  2. In section 3.2 it states that thirty-three children with WT required unplanned PICU admission. However, in Figure 1 it states that there were 50 unplanned PICU admissions analyzed in this manuscript. When I count the clinical characteristics rows in Table 2, I come up with 43 patients. Please account for this discrepancy and clarify how the same patient having multiple PICU admissions was accounted for in the analysis. I think the asterixes in table 2 describe this, but I would also add this more clearly and completely in the text.  
  3. Section 3.2 states that patients with germline WT1 alteration more frequently had unplanned PICU admission. Were there certain reasons this group was admitted to the PICU? Kidney disease? Otherwise? 
  4. Was there any obvious difference in PICU admission patterns after January 2015 when all cases were diagnosed in treated in the national pediatric oncology center compared to the earlier study period?
  5. I was interested to see that subglottic stenosis was a recurrent reason for unplanned PICU admission. Please add a sentence to the discussion about this. I have never seen this in a patient with Wilms tumor - looks like one of the patients in this series was a neonate which makes sense. what about the other one? The other reasons for unplanned PICU admission were more along expected lines. 

Minor:

  1. Italicize WT1 when referring to the gene throughout the manuscript.
  2. Line 145 "In case blood pressure measurements were not performed..." Please change to "If blood pressure measurements were not performed..."
  3. Line 169 "Of six patients..." please change to "For six patients..."
